# Metabolomics-Based Analysis of the Effects of Different Cultivation Strategies on Metabolites of *Dendrobium officinale* Kimura et Migo

**DOI:** 10.3390/metabo13030389

**Published:** 2023-03-06

**Authors:** Da Yang, Yeyang Song, Anjin Lu, Lin Qin, Daopeng Tan, Qianru Zhang, Yuqi He, Yanliu Lu

**Affiliations:** 1Key Lab of the Basic Pharmacology of the Ministry of Education, Zunyi Medical University, 6 West Xue-Fu Road, Zunyi 563009, China; 2Guizhou Engineering Research Center of Industrial Key-technology for Dendrobium Nobile, Zunyi Medical University, 6 West Xue-Fu Road, Zunyi 563000, China

**Keywords:** *Dendrobium officinale* Kimura et Migo, different cultivation strategies, metabolites, metabolomics, ultra-performance liquid chromatography, tandem mass spectrometry

## Abstract

*Dendrobium officinale* Kimura et Migo is a famous plant with a high medicinal value which has been recorded in the Chinese Pharmacopoeia (2020 Edition). The medicinal properties of *D. officinale* are based on its chemical composition. However, there are no reports on how different cultivation methods affect its chemical composition. In order to reveal this issue, samples of the *D. officinale* were collected in this study through tree epiphytic cultivation, stone epiphytic cultivation, and greenhouse cultivation. Polysaccharides were determined by phenol sulfuric acid method and secondary metabolites were detected by the UPLC-MS technique. In addition, with regards to metabolomics, we used multivariate analyses including principal component analysis (PCA) and orthogonal partial least squares analysis (OPLS-DA) to screen for differential metabolites which met the conditions of variable importance projection values >1, fold change >4, and *p* < 0.05. The differential metabolites were taken further for metabolic pathway enrichment analysis, which was based on the Kyoto Encyclopedia of Genes and Genomes (KEGG) database, and validated by antioxidant activity. Comparing the three groups of samples according to the standards of the ChP (2020 edition), the results showed that the polysaccharide content of the samples from stony epiphytic cultivation and greenhouse cultivation was significantly higher than that of the samples from live tree epiphytic cultivation. Metabolomic analysis revealed that there were 185 differential metabolites among the 3 cultivation methods, with 99 of the differential metabolites being highest in the stone epiphytic cultivation. The results of the metabolic pathway enrichment analysis showed that the different cultivation strategies mainly effected four carbohydrate metabolic pathways, five secondary metabolite synthesis pathways, six amino acid metabolic pathways, one nucleotide metabolism pathway, three cofactor and vitamin metabolism pathways, and one translation pathway in genetic information processing. Furthermore, *D. officinale* from stone epiphytic cultivation which had the best antioxidant activity was implicated in differential metabolite production. This study revealed the effects of different cultivation methods on the chemical composition of *D. officinale* and also provided a reference for establishing the quality control standards to aid its development and utilization.

## 1. Introduction

*Dendrobium officinale* Kimura et Migo is a famous plant with high medicinal value which has been recorded in Chinese Pharmacopoeia due to its various remarkable effects, such as cardiovascular protection [1,2], hypoglycemia [3,4], gastrointestinal protection [5,6,7], immune modulation [8,9], antitumor [10,11,12], anti-aging [13,14], and anti-osteoporosis [15]. Its main chemical constituents include polysaccharides [16], flavonoids [17], phenolic acids [18], amino acids [19], bibenzyl [20], alkaloids [21], coumarins, lignans [22], and organic acids [23,24], which are closely related to its pharmacological effects. (Specific information on some of the Dendrobium candidum compounds reported in the literature is shown in Appendix A). Many studies have reported the pharmacological activities of *D. officinale*; however, few reports have considered the secondary metabolites or biological activities of *D. officinale* from the perspective of the whole plant system.

*D. officinale* is mainly distributed south of the Yangtze River. Some studies suggest that it has a long history of application in the Yunnan-Guizhou Plateau, with its northward or eastward migration forming the first iteration of expansion [25]. The scarcity of wild resources [26] and the huge market demand led to the emergence of its artificial cultivation [27]. Since the 21st century, the cultivation of *D. officinale* has made significant progress, mainly through wild-like cultivation and greenhouse cultivation. The wild-like cultivation is further divided into live tree epiphytic cultivation and stone epiphytic cultivation [28,29]. This has the advantages of making full use of natural resources, having a low cost of cultivation, and having little difficulty in the management and care; greenhouse cultivation adds more artificial cultivation than wild cultivation, allowing it to grow in the greenhouse with the most suitable growth habits and preventing the occurrence of diseases as much as possible [27].

It is well known that the composition and content of chemical components of plants are influenced by their growth environment [30,31]. The content and variation of secondary metabolites in plants directly affects their quality differences and activity [32]. For this reason, many studies have also aimed to enhance the pharmacological activity of plants by altering their growth environment to increase the content of secondary metabolites [33,34]. Currently, commercially circulating *D. officinale* is mainly stone epiphytic cultivation, live tree epiphytic cultivation, and greenhouse cultivars [28,29] However, only the plant appearance of *D. officinale* has been reported to vary under different cultivation methods [35], and the effects of different cultivation methods on the chemical composition have not been compared. Therefore, in order to ensure the stable quality of *D. officinale*, it is necessary to carry out studies related to the differences in metabolites and the processes affecting *D. officinale* in different cultivation methods. 

In the present study, the samples of *D. officinale* from the three cultivation strategies, including live tree epiphytic cultivation, stone epiphytic cultivation, and greenhouse cultivation, were collected. The polysaccharide content determination was conducted by the phenol-sulphuric acid method and the metabolomic study conducted by UPLC-MS technology. The antioxidant activity of different cultivation methods was verified by DPPH and ABTS+ free radical scavenging assays. The results could also inform the establishment of quality control standards to aid in its development and utilization.

## 2. Results

### 2.1. Effects of Different Cultivation Methods on the Polysaccharide Content of D. officinale 

The effects of different cultivation methods on the polysaccharide content of *D. officinale* were shown in Figure 1. The polysaccharide content of *D. officinale* cultivated epiphytically on trees was significantly lower than that of those cultivated in the other 2 ways (*p* < 0.05). There was no statistical difference in the polysaccharide content between samples from stone epiphytic cultivation and greenhouse cultivation. According to the Chinese Pharmacopoeia 2020 Edition, the polysaccharide content of *D. officinale* should not be less than 25%. The samples from stone epiphytic cultivation and greenhouse cultivation had the same qualification rate of polysaccharide content, both at 66.6%.

### 2.2. Metabolomic Determination of D. officinale by UPLC-MS 

We selected 18 representative samples of *D. officinale*. This study was conducted by a UPLC-MS system, which was used to determine the metabolites of the samples in the positive and negative ion mode, respectively. The quality control information is shown in Appendix A. The results showed that a total of 949 metabolites were detected. As shown in Figure 2, the metabolites were analyzed and classified, including 177 phenolic acids, 153 lipids, 121 flavonoids, 86 amino acids and their derivatives, 83 organic acids, 66 alkaloids, 55 nucleotides and their derivatives, 28 lignans and coumarins, 27 terpenoids, 24 quinones, 16 vitamins, 7 stilbenes, 70 saccharides and alcohols, and 36 other compounds. 

### 2.3. Effects of Different Cultivation Methods on the Metabolite Profile of D. officinale 

Unsupervised pattern recognition principal component analysis (PCA) was used to compare whether the secondary metabolites of *D. officinale* were affected under different cultivation conditions from the profiles. As shown in Figure 3, the points representing the samples from the three cultivation strategies were significantly distinguished in the PC1 and PC2 directions and did not intersect with each other. This study suggested that the three different cultivation methods had significant effects on the metabolite profile of *D. officinale*.

### 2.4. Effects of Different Cultivation Methods on the Metabolites of D. officinale

Orthogonal partial least squares analysis (OPLS-DA) was used to compare and find out the different metabolites of *D. officinale* under the three cultivation conditions. As can be seen from Figure 4A–C, the OPLS-DA analysis models were stable and valid, and the variable importance projection values (VIP values) generated by the models were also reliable. The points representing the samples from the different cultivation strategies were clearly distinguished. In order to screen the most representative variable metabolites, the criteria were set to satisfy the conditions of VIP value >1, fold change >4, and *p* < 0.05. Compared with the samples from stone epiphytic cultivation, the contents of 103 metabolites in the samples from greenhouse cultivation changed significantly, as shown in Figure 4D. There was an increase in the contents of 8 metabolites and a decrease in the contents of 95 metabolites. Compared with the samples from live tree epiphytic cultivation, the contents of 65 metabolites in the samples from greenhouse cultivation changed significantly, as shown in Figure 4E. There was an increase in the contents of 19 metabolites and a decrease in the contents of 46 metabolites. Compared with the samples from stone epiphytic cultivation, the contents of 117 metabolites in the samples from live tree epiphytic cultivation changed significantly, as shown in Figure 4F. There was an increase in the contents of 36 metabolites and a decrease in the contents of 81 metabolites. Through analysis, it was found that the changed metabolites included phenolic acids, flavonoids, lipids, amino acids and their derivatives, organic acids, alkaloids, nucleotides and their derivatives, lignans, coumarins, terpenoids, quinones, others, and so on. The details are shown in Table 1.

There were 185 metabolites that were significantly altered in the samples from the 3 incubation methods after de-weighting. Among the metabolites, the contents of 99 metabolites were the highest under stone epiphytic cultivation, the contents of 71 metabolites were the highest under live tree epiphytic cultivation, and the contents of 15 metabolites were the highest under greenhouse cultivation, as shown in Figure 5. The content of secondary metabolites of the different species is significantly higher in stone epiphytic cultivation. Live tree epiphytic cultivation is the second most abundant, less so than in greenhouse cultivation. (Detailed information is shown in Appendix A.)

### 2.5. Effects of Different Cultivation Methods on the Metabolic Pathways of D. officinale

Metabolites were annotated and imported into Metabo Analyst 5.0 for KEGG metabolic pathway enrichment analysis. The top twenty enriched metabolic pathways were presented as enrichment bubble plots, as shown in Figure 6. Within these pathways, there are four carbohydrate metabolic pathways, five secondary metabolite synthesis pathways, six amino acid metabolic pathways, one nucleotide metabolism pathway, three cofactor and vitamin metabolism pathways, and one translation pathway in genetic information processing. The flavone and flavonol biosynthesis pathways were affected most by the three cultivation methods due to the highest degree of enrichment and the greatest number of changed metabolites. 

### 2.6. Effects of Different Cultivation Methods on the Antioxidant Activity of D. officinale

The metabolic pathways with the highest enrichment degree were the flavone and flavonol biosynthesis pathway and the ascorbic acid and aldose metabolism pathway, which were known to have powerful antioxidant capacities. Therefore, the antioxidant activity of *D. officinale* under different cultivation methods was verified. As shown in Figure 7, all samples showed a dose-dependent relationship in terms of DPPH and ABTS radical scavenging rates. However, the *D. officinale* from stone epiphytic cultivation showed higher DPPH and ABTS^+^ radical scavenging ability than the other two culture methods at the same concentration.

## 3. Discussion

As mentioned before, *D. officinale* is a plant with a high value in clinical applications. The material basis of its multiple pharmacological activities is the various chemical compounds it contains. We demonstrate that the cultivation method influences the metabolite composition. Chemical compounds could be affected by the different cultivation methods. For example, the polysaccharide content of *D. officinale* from greenhouse cultivation and stone epiphytic cultivation was higher than that of *D. officinale* from live tree epiphytic cultivation. In addition, 185 metabolites were significantly altered by the different cultivation methods. Among the metabolites, the contents of 99 metabolites were the highest under stone epiphytic cultivation, the contents of 71 metabolites were the highest under live tree epiphytic cultivation, and the contents of 15 metabolites were the highest under greenhouse cultivation. The metabolites mainly included phenolic acids, flavonoids, lipids, amino acids, organic acids, alkaloids, and so on. 

The differences in metabolite content between cultivation conditions may be due to the plants’ self-regulation. In order to reveal the possible mechanism of the effects, metabolic pathway enrichment analysis was performed. The top twenty enriched metabolic pathways were found. The flavone and flavonol biosynthesis pathway was most affected by the three cultivation methods. Through this pathway, flavonoids and flavonols would be converted into lignin, which would accumulate in the secondary cell walls of plants, participate in mechanical support, and form conduits for transporting water and mineral elements, thus improving the drought tolerance of plants [36]. There were other pathways among the 20 pathways that can help plants with drought resistance. Lignin is also produced through the phenylpropanoid pathway, and thus the corresponding effects can be found [37]. Arginine had been reported to prevent water deficit-induced accumulation of proline, improve leaf gas exchange during water deficit, and enhance root antioxidant capacity in recovering plants, contributing to plant growth and development in water-stressed environments [38]. In the present study, the enrichment of arginine biosynthesis was most likely due to the self-regulation of *D. officinale* to improve the water deficit in stone and live tree epiphytic cultivation. In addition, β-alanine had multiple functions for plants. The β-alanine metabolic pathway was involved in protecting plants from extreme temperature, drought, hypoxia, heavy metal shock, and some biological stresses [39]. The self-regulation of *D. officinale* was manifested not only in drought resistance but also in other aspects. For example, the ascorbate and aldose metabolism pathway would help to regulate the plant’s own redox balance to adapt to nutrient-deficient survival conditions [40]. These results suggested that, under wild-like cultivation, *D. officinale* enhanced its ability to cope with the complex growth environment by instinctively self-regulating the levels of metabolites. 

In addition, different cultivation methods have the potential to affect the pharmacological activity of *D. officinale* by changing the metabolites levels. The polysaccharide of *D. officinale* has been reported to have a wide range of pharmacological activities such as antitumor [10], immunomodulatory [8], and hypoglycemic [2]. Compared to live-tree epiphytic cultivation, *D. officinale* from stone epiphytic cultivation and greenhouse cultivation contained more polysaccharide content with statistical-mathematical significance. The flavone and flavonol biosynthesis and the ascorbate and aldarate metabolism pathways, whose metabolites have important antioxidant effects [41,42],were the most variable among the metabolic pathways. To further compare the effect of these differential metabolites of *D. officinale* under three cultivation methods, we used DPPH and ABTS scavenging activities to analyze the antioxidant activity [43], while ascorbic acid (VitC) has been used as a positive agent in free radical scavenging assays [44]. *D. officinale* from stone epiphytic cultivation showed the best antioxidant ability than the other two culture methods at the same concentration, and the greenhouse cultivation showed the weakest. In the two pathways, six compounds underwent significant changes. Four of these compounds (Kaempferol 3-O-glucoside, D-Glucuronate, Dehydroascorbate, and D-Glucurono-6, 3-lactone) were most abundant in *D. officinale* from stone epiphytic cultivation. In addition, two of these compounds (Kaempferol and Quercetin-3-O-rutinoside) were higher in *D. officinale* from live-tree epiphytic cultivation. These results suggested that different cultivation methods do have a significant effect on the chemical composition of *D. officinale* and, therefore, on its biological activity.

As mentioned earlier, there were many compounds that were the most abundant and had important pharmacological effects in *D. officinale* from stone epiphytic cultivation. For example, hircinol has been found to have antiproliferative and apoptosis-inducing effects on gastric cancer cell lines and was considered a promising candidate for drugs and nutrients [45]. Lumichrome has been reported to inhibit osteoclastogenesis and bone resorption by inhibiting RANKL-induced NFAT activation and calcium signaling, suggesting that lumichrome has potential as a therapeutic agent for osteolytic diseases [46]. The anti-inflammatory effects of rutinoside have been reported [47], and in recent studies its combination with ascorbic acid has been found to be effective at the treatment of pigmented purpuric skin disease [48]. Taxifolin has also been shown to have biological activity against a variety of cancers, such as osteosarcoma [49], colorectal cancer [50], breast cancer [51], and lung cancer [52]. 

In conclusion, the value of medicinal plants has been often influenced by their secondary metabolites, the production of which was inextricably linked to growth and development and environmental factors [53]. The results of this experiment showed that there are significant effects on the chemical composition of *D. officinale* due to differences in the growing environment. *D. officinale* attached to stones epiphytic cultivated with more flavonoid and phenolic acid metabolites, which enhanced its antioxidant capacity and ability to cope with the complex survival environment. We suggest that the choice of cultivation method for *D. officinale* is very important. Furthermore, we also suggest that the quality standard of *D. officinale* needs to consider not only the polysaccharide content, but also for the content of other compounds.

## 4. Materials and Methods

### 4.1. Collection Information of D. officinale Samples

Fresh stems were collected from the Anlong and Xingyi areas of Guizhou Province, China. (Photographs of the different cultivation methods of Dendrobium are shown in Appendix A).The collected samples were identified as *D. officinale* by Professor Jianwen Yang of Zunyi Medical University. Details of the sample collection are shown in Table 2.

### 4.2. Instruments and Reagents

The instruments used were as follows: SHIMADZU Nexera X2 UPLC (Kyoto, Shimadzu, Japan), Applied Biosystems 4500 QTRAP mass spectrometer (Applied Biosystems, Waltham, MA, USA), MULTISKAN GO full-wavelength microplate reader (Thermo Fisher, Waltham, MA, USA), 3K15 centrifuge (Sigma-Aldrich, St. Louis, MO, USA), WP-UP-YJ-20 micro-organic heat removal ultra-pure water machine (Sichuan Water Treatment Equipment Co., Chengdu, China), and ME204E electronic balance (Shanghai Mettler-Toledo Instruments, Shanghai, China). The reagents used were as follows: 2,2-biphenyl-1-bitter hydrazinyl (Shanghai Macklin Biochemical Technology Co., Ltd., Shanghai, China), 2,2′-Hydrazinyl-bis(3-ethylbenzothiazoline-6-sulphonic acid) diamine salt (Shanghai Macklin Biochemical Technology Co., Ltd., Shanghai, China), glucose (Sigma-Aldrich, St. Louis, MO, USA), anhydrous ethanol (analytical grade, Chengdu Kelong Chemical Reagent Factory, Chengdu, China), phenol (analytical grade, Chengdu Kelong Chemical Reagent Factory, Chengdu, China), concentrated sulphuric acid (analytical grade, Sinopharm Chemical Reagent Co., Ltd., Shanghai, China), acetonitrile (mass spectrometry grade, Merck & Co., Rahway, NJ, USA), and methanol (mass spectrometry grade, Merck & Co., Rahway, NJ, USA).

### 4.3. Determination of Polysaccharide Content of D. officinale 

#### 4.3.1. Extraction of Polysaccharide

D. officinale was dried at 60 °C, crushed, and sieved through 50-mesh sieve to obtain the powder. The powder was precisely weighed to 60 mg and refluxed with 40 mL of water for 2 h. The solution was cooled and the volume fixed with water at 50 mL, mixed well, and centrifuged at 4000 rpm for 15 min. In total, 2.0 mL of the supernatant was mixed well with 10 mL of anhydrous ethanol and placed at 4 °C for 1 h, then centrifuged at 4000 rpm for 20 min. The precipitate was washed 2 times with 8 mL of 80% (*v*/*v*) ethanol solution and centrifuged at 4000 rpm for 20 min. The sample was obtained by dissolving the precipitate using hot water and fixing its volume at 10 mL. 

#### 4.3.2. Polysaccharide Assay

In total, 1.0 mL of 5% phenol solution and 5.0 mL of concentrated sulfuric acid were added to 1.0 mL of the sample. The reaction solution was mixed and heated in water at 100 °C for 20 min, then put in an ice bath for 5 min. The absorbance of the reaction solution was measured at 488 nm using a microplate reader.

#### 4.3.3. Preparation of Glucose Standard Curve

In total, 9.00 mg of glucose standard was dissolved in water and the volume fixed to 50 mL to obtain the stock solution (containing 180 μg/mL of glucose). The stock solution was diluted to 0, 30, 60, 90, 120, and 150 μg/mL to obtain the standard solution. 

### 4.4. Determination of the Metabolomics of D. officinale by UPLC -MS/MS 

The analysis was performed on the Agilent SB-C18 column (2.1 × 100 mm^2^, 1.6 μm). The mobile phase consisted of ultra-pure water (containing 0.1% formic acid) for phase A and acetonitrile (containing 0.1% formic acid) for phase B. The elution gradient was as follows: 0–9 min, 5–95% for phase B; and 9–10 min, 95% for phase B. The flow rate was 0.35 mL/min, the column temperature was 40 °C, and the injection volume was 4 μL. The parameters of the ESI source were as follows: the source temperature was set to 550 °C; the ion spray voltage was set to 5500 V for the positive ion mode and −4500 V for the negative ion mode; the pressures of ion source gas I, gas II, and curtain gas were set to 50, 60, and 25.0 psi, respectively; and the collision-induced ionization parameter was set to high. Further DP and CE optimization was completed for individual MRM ion pairs. In total, 100 mg of *D. officinale* powder was added to 1.2 mL of 70% (*v*/*v*) methanol and vortexed every 30 min for 30 s for 6 times. The samples were placed overnight at 4 °C and centrifuged for 10 min at 12,000 rpm. The supernatant was filtered through a microporous membrane (0.22 μm). The filtrate was used for UPLC-ESI-MS/MS analysis.

The identification of the chemical composition was based on the MWDB (metware database, http://en.metware.cn/list/27.html (accessed on 26 December 2021)), which is a self-built database from Wuhan, China. The characterization of the substance was performed based on the secondary spectral information with the removal of isotopic signals and repeating signals. The quantification was achieved using triple quadrupole mass spectrometry in multiple reaction monitoring (MRM) mode. The mass spectra were integrated and calibrated using MultiQuant MD 3.0.3 software and the peak area represented the relative content of the corresponding substance [33].

### 4.5. KEGG Metabolic Pathway Enrichment

Based on the KEGG compound database, the metabolites were annotated. Furthermore, the annotated metabolites were then matched based on the KEGG pathway database. Significantly regulated metabolites were imported into the metabolite enrichment analysis MetaboAnalyst 5.0 for their metabolic pathway enrichment. The enrichment results were calculated for the significance (p-values) by hypergeometric tests and presented as bubble plots.

### 4.6. Comparing the Antioxidant Properties of D. officinale Samples

#### 4.6.1. Sample Extraction 

In total, 0.9 g of *D. officinale* powder was added to 90 mL of 70% (*v*/*v*) ethanol, extracted at reflux for 2 h, made up to the weight, filtered, and concentrated to 25 mg/mL. The concentrates were diluted to 0, 5, 10, 15, 20, and 25 mg/mL, and stored at 4 °C.

#### 4.6.2. DPPH Radical Scavenging Assay

In total, 0.5 mL of the sample solution was added to 1.75 mL of 1.9 × 10^−4^ mol/L DPPH solution, mixed immediately, and reacted for 30 min at room temperature and protected from light. Ascorbic acid (Vit C) with the same concentration as the sample was used as a positive control. The absorbance was measured at 517 nm. The clearance of DPPH (%) = [1 − (A − A_1_)/A_0_] × 100% was calculated to compare the antioxidant activity between the different cultivation methods. A was the absorbance of the sample. A_1_ was the absorbance of the sample with the same volume of anhydrous ethanol as the DPPH solution. A_0_ was the absorbance of the same volume of 70% (*v*/*v*) ethanol as the sample with the DPPH solution.

#### 4.6.3. ABTS Radical Scavenging Assay

ABTS free radical solution (ABTS^+^) was prepared by mixing 7 mol/L of ABTS solution and 2.45 mol/L of potassium persulfate solution at 1:1, which was then placed at room temperature and protected from light for 12 h before use. The solution was diluted with PBS (pH 7.4) until the absorbance at 734 nm was 0.70 (±0.02). In total, 0.2 mL of the sample solution was added to 5 mL of ABTS^+^ solution, mixed thoroughly, and reacted for 6 min at room temperature. Ascorbic acid (VitC) with the same concentration as the sample was used as the positive control. The absorbance at 734 nm was recorded immediately. The clearance of ABTS^+^ (%) was calculated = [1 − (A − A_1_)/A_0_] × 100% to compare the antioxidant activity between the different cultivation methods. A was the absorbance of the sample. A_1_ was the absorbance of the sample with the same volume of distilled water as ABTS^+^ solution. A_0_ was the absorbance of the same volume of 70% (*v*/*v*) ethanol as the sample with ABTS^+^ solution.

### 4.7. Data Analysis 

All visualization charts in the paper were performed in R programs. The FactoMineR package was used for PCA, the ropls package was used for OPLS-DA, the mixOmics package was used for volcano plots, and the other packages (ggsignif, gridExtra, reshape2 showtext, ggrepel, etc.) were also used. For multiple group comparisons, one-way analysis of variance (ANOVA) was performed using SPSS 18.0 (IBM, Chicago, IL, USA). *p* < 0.05 was considered statistically significant.

## 5. Conclusions

In this study, the effects of three cultivation methods on the chemical composition of *D. officinale* were analyzed by comparing the polysaccharide content and the types of secondary metabolites in live tree epiphytic cultivation, stone epiphytic cultivation, and greenhouse cultivation for *D. officinale* and their intrinsic influencing factors. The results showed that the polysaccharide contents of *D. officinale* were comparable and significantly higher in stone epiphytic cultivation and greenhouse cultivation than in live tree epiphytic cultivation. The results of the secondary metabolite analysis showed that the secondary metabolites were significantly higher in the stone epiphytic cultivation *D. officinale* than in the live tree epiphytic cultivation and greenhouse cultivation. The content of flavonoids and phenolic acid was significantly higher in *D. officinale* stone epiphytic cultivation than in live tree epiphytic cultivation and greenhouse cultivation. The enrichment of the metabolic pathways in *D. officinale* indicates that there are differences in the synthesis of major flavonoid biosynthesis between cultivation methods. In addition, the results of antioxidant activity further demonstrated that the antioxidant activity of *D. officinale* stone epiphytic cultivation was significantly stronger than other cultivation methods. This study shows that the different cultivation methods of *D. officinale* have a great influence on its chemical composition. It also provides a reference for the selection of high-quality *D. officinale* cultivation methods and the selection of raw materials for product development. 

## Figures and Tables

**Figure 1 metabolites-13-00389-f001:**
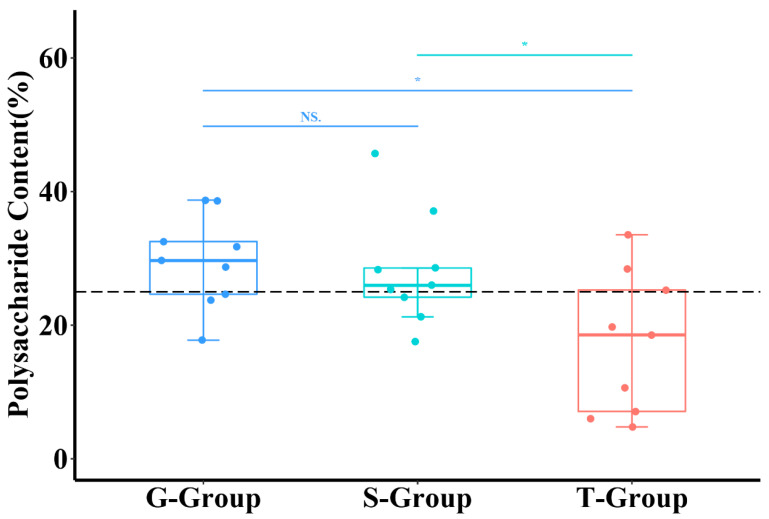
The effects of different cultivation methods on polysaccharide content (*n* = 9). Each point represented one sample. The G-Group meant that the sample was from greenhouse cultivation; S-Group meant that the sample was from stone epiphytic cultivation; T-Group meant that the sample was from live tree epiphytic cultivation. The dotted line indicated the polysaccharide content was equal to 25%. Furthermore, * represented *p* < 0.05, and NS represented that there was no statistical difference.

**Figure 2 metabolites-13-00389-f002:**
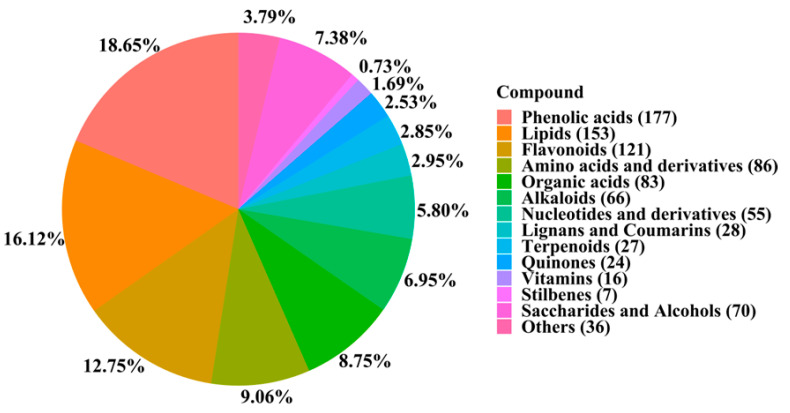
The classification and of the metabolites. This pie chart depicts the proportion of different types of metabolites among the metabolites detected by UPLC-MS.

**Figure 3 metabolites-13-00389-f003:**
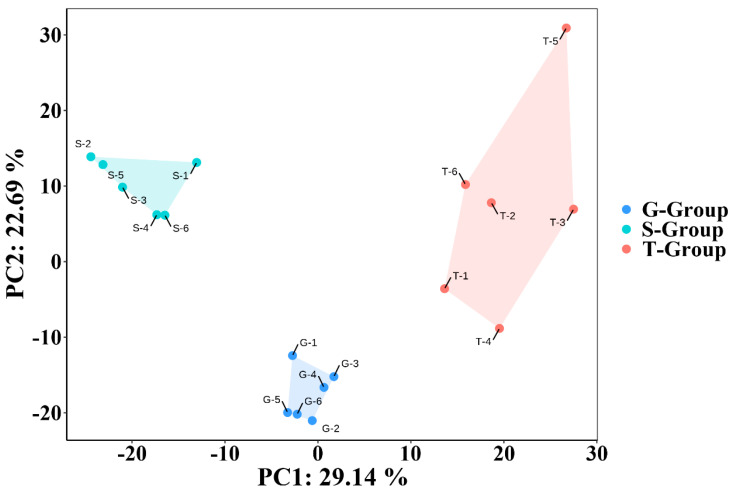
The effects of different cultivation methods on the metabolite profile (*n* = 6). PCA was used to analyze the data. The contribution of PC1 was 29.14% and that of PC2 was 22.69%. Each point represented one sample. G-Group meant that the sample was from greenhouse cultivation; S-Group meant that the sample was from stone epiphytic cultivation; T-Group meant that the sample was from live tree epiphytic cultivation.

**Figure 4 metabolites-13-00389-f004:**
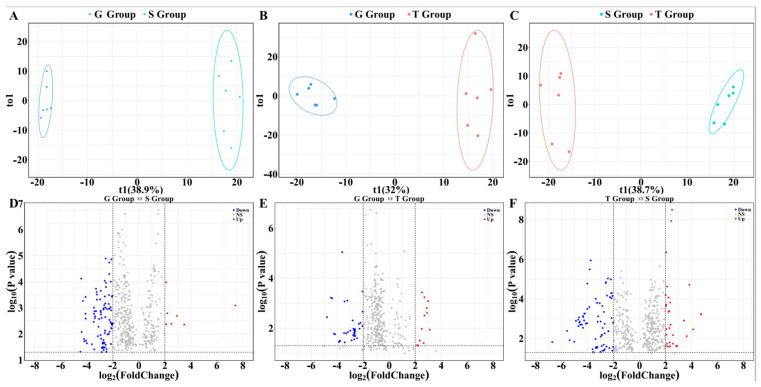
The effects of different cultivation methods on the metabolites (*n* = 6). A supervised OPLS-DA model was used to compare the data. The G-Group meant that the sample was from greenhouse cultivation; S-Group meant that the sample was from stone epiphytic cultivation; T-Group meant that the sample was from live tree epiphytic cultivation. In (**A**–**C**), the x-axis represents the predicted principal component and the y-axis represents the orthogonal principal component. Each point represents one sample. (**A**) compares the samples from greenhouse cultivation and stone epiphytic cultivation with parameters R^2^X: 0.483, R^2^Y: 0.997, and Q^2^: 0.93. (**B**) compares the samples from greenhouse cultivation and live tree epiphytic cultivation with parameters R^2^X: 0.527, R^2^Y: 0.986, and Q^2^: 0.916. (**C**) compares the samples from stone epiphytic cultivation and live tree epiphytic cultivation with parameters R^2^X: 0.511, R^2^Y 0.991, and Q^2^: 0.936. (**D**–**F**) includes the volcano plots which were used to show the differences in metabolites between the groups. The screening criteria were set to satisfy the conditions of VIP value >1, fold change >4, and *p* < 0.05. The blue dots in the graphs indicate that the metabolite contents were significantly down-regulated. The red dots indicate that the metabolite contents were significantly up-regulated. The grey dots indicated that the metabolite contents were not significantly changed.

**Figure 5 metabolites-13-00389-f005:**
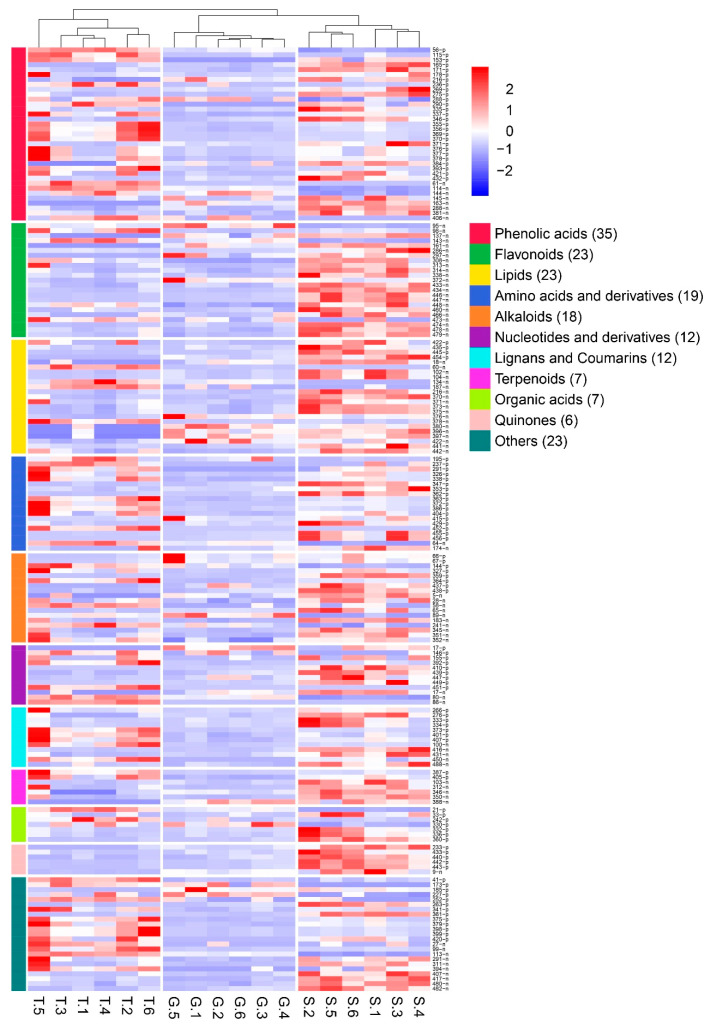
The effects of different cultivation methods on the contents of the changed metabolites (*n* = 6). The heat map was used to describe the changes in the metabolite contents. G meant that the sample was from greenhouse cultivation; S meant that the sample was from stone epiphytic cultivation; T meant that the sample was from live tree epiphytic cultivation. On the left side of the graph, the color was used to show the classification of the metabolites. On the right side of the graph, the number of the metabolite was listed. N meant that the determination was in negative ion mode, and P meant that the determination was in positive ion mode. Re-ID is a ranking of compounds based on their molecular weight and the results of positive and negative ion scanning patterns. In the graph, the red color represents an increase in metabolite content and the blue color represents a decrease in metabolite content.

**Figure 6 metabolites-13-00389-f006:**
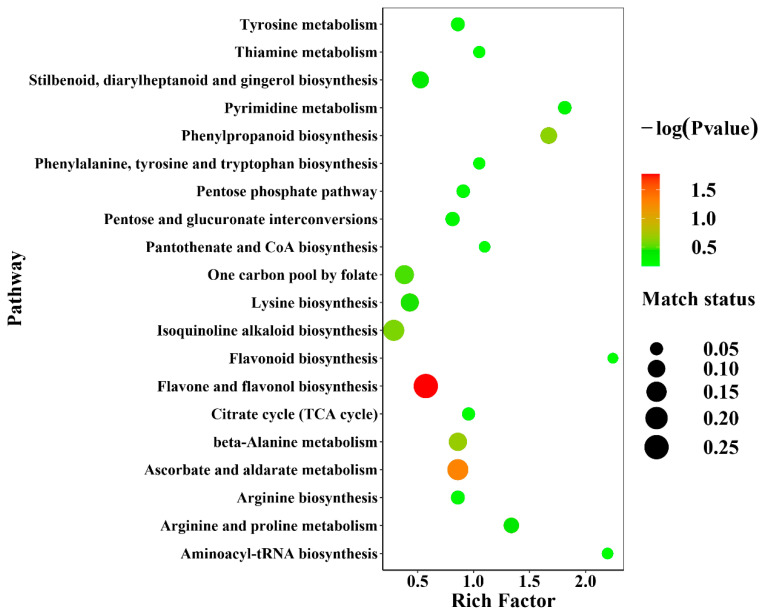
The effects of different cultivation methods on the metabolic pathways. The bubble diagram was used to show the results of KEGG metabolic pathway enrichment analysis. On the left side of the graph, the names of the top twenty enriched metabolic pathways are labeled. The rich factor indicates the degree of enrichment. The color of the dots represents the *p*-value and the size of the dots represents the amount of metabolites enriched in the pathway.

**Figure 7 metabolites-13-00389-f007:**
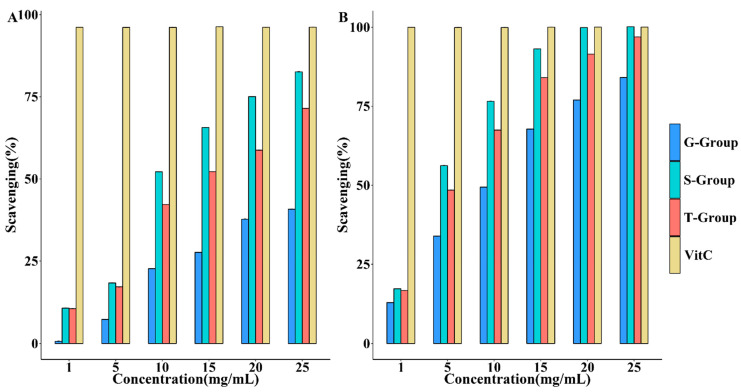
The effects of different cultivation methods on the antioxidant activity of *D. officinale.* (**A**) describes DPPH scavenging capacity and (**B**) describes ABTS^+^ scavenging capacity. G-Group means that the sample was from greenhouse cultivation; S-Group means that the sample was from stone epiphytic cultivation; T-Group means that the sample was from live tree epiphytic cultivation.

**Table 1 metabolites-13-00389-t001:** Classification of the different metabolites.

Category	G vs. S		G vs. T		T vs. S	
	UP	DOWN	UP	DOWN	UP	DOWN
Phenolic acid	2	15	2	12	10	13
Flavonoids	1	15	3	1	3	10
Lipids	1	13	6	3	3	14
Alkaloids	1	8	1	3	4	7
Saccharides and alcohols	2	5	0	3	2	4
Amino acids and their derivatives	0	10	2	7	3	7
Organic acids	1	4	0	0	2	3
Lignans and coumarins	0	7	0	5	2	5
Others	0	3	0	4	1	2
Nucleotides and their derivatives	0	4	2	3	4	6
Quinones	0	5	2	0	0	6
Terpenoids	0	4	1	2	0	4
Vitamins	0	1	0	2	2	0
Stilbenes	0	1	0	1	0	0
Total	8	95	19	46	36	81

G meant that the sample was from greenhouse cultivation; S meant that the sample was from stone epiphytic cultivation; T meant that the sample was from live tree epiphytic cultivation. UP signifies the number of metabolites with increased content and DOWN signifies the number of metabolites with decreased content.

**Table 2 metabolites-13-00389-t002:** The information of the selected samples.

Growth Method	Epiphytic Substrates	Time of Harvesting	Growing Years	Elevation (m)	Climate Type	Lightwardness	Longitude and Latitude	Number of Samples	Place of Origin
Tree epiphytic cultivation	Silktree Siris	201906	3 years old	1122	Subtropical monsoon climate	Light	E105°27′57”N24°54′3”	9	Anlong
Stone epiphytic cultivation	Limestone	201906	3 years old	1172	Subtropical monsoon climate	Light	E104°57′54”N24°54′18”	9	Xingyi
Greenhouse cultivation	Wood chips	201906	3 years old	1160	-	-	E104°57′49”N24°54′17”	9	Xingyi

## Data Availability

The datasets used during the current study are not publicly available due to participant confidentiality issues. They are available to corresponding authors upon reasonable request.

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
