# Peer review of "Metabolomics-Based Analysis of the Effects of Different Cultivation Strategies on Metabolites of Dendrobium officinale Kimura et Migo"

_metabolites, 2023, doi:10.3390/metabo13030389_

Round 1

Reviewer 1 Report

The manuscript presents the change evaluation in terms of the chemical composition of D. officinale, grown in three different substrates. The main results obtained with metabolomic analysis, reveal that the substrate of cultivation influences the chemical composition of plant biomass, and most probably affects the metabolic pathways. The study tries to put into evidence the effects on the chemical composition of D. officinale, as a result of the influences of the different substrates,  such as live tree epiphytic, stone epiphytic and greenhouse cultivation,  and this represents in fact the novelty of the article. The manuscript is well-written, clear and concise and the methodology used is correct. However, before publishing the authors must read with attention the manuscript, because:

 1) there are some type-written mistakes;

 2)  some sentences must be revised  (i.e sentence from rows 301-302; the last part of the sentence from row 305).

Author Response

Manuscript ID: metabolites-2188193

Title: Metabolomics-based analysis of the effects of different cultivation
strategies on metabolites of Dendrobium officinale Kimura et Migo

Dear Reviewer ,

Thank you very much for your email. We have carefully revised the manuscript according to your suggestions. Enclosed is the point-by-point response to these comments, along with the Highlights sections of the revised manuscript. We hope that with these reversions, this manuscript is suitable for publication in your journal. We thank the reviewers for their scientific perspectives.

I am looking forward to hearing from you sooner.

Sincerely yours,

Da Yang

Key Lab of the Basic Pharmacology of the Ministry of Education, Zunyi Medical
University,6 West Xue-Fu Road,Zunyi City, Guizhou

E-Mail: [email protected]

February 8, 2023

  1. There are some type-written mistakes;

Response: We thank the reviewer for the comment. We tried our best to improve the manuscript and correcting mistakes. These changes will not influence the content and framework of the paper.

  1. Some sentences must be revised (i.e sentence from rows 301-302; the last part of the sentence from row 305).

Response: We thank the reviewer for the comment. As suggested by the reviewer, we have corrected the “The precipitate was dissolved in hot water and fixed the volume to 10 mL. The sample was obtained.” into “The sample was obtained by dissolving the precipitate using hot water and fixing its volume at 10 mL.”. We have corrected the “The absorbance of the reaction solution was measured at 488 nm.” into “Absorbance of the reaction solution was measured at 488nm using microplate reader”.

Reviewer 2 Report

The article by Yang et al. describes the analysis of the effects of different cultivation strategies on metabolites of Dendrobium officinale.

It is a valuable paper but need some changes.

It would be clearer to put the composition of verses 48-50 in the table with examples of specific compounds.

Is it possible to show pictures of different cultivars of the plant?

Please write about previous composition studies, e.g. Chen, W. H., Wu, J. J., Li, X. F., Lu, J. M., Wu, W., Sun, Y. Q., ... & Qin, L. P. (2021). Isolation, structural properties, bioactivities of polysaccharides from Dendrobium officinale Kimura et. Migo: A review. International Journal of Biological Macromolecules, 184, 1000-1013.

Jin, Q., Jiao, C., Sun, S., Song, C., Cai, Y., Lin, Y., ... & Zhu, Y. (2016). Metabolic analysis of medicinal Dendrobium officinale and Dendrobium huoshanense during different growth years. PLoS One, 11(1), e0146607.

Yuan, H., Bai, Y., Si, J., Zhang, A., & Jin, X. (2011). Variation of monosacchride composition of polysacchrides in Dendrobium officinale by pre-column derivatization HPLC method. Zhongguo Zhong yao za zhi= Zhongguo Zhongyao Zazhi= China Journal of Chinese Materia Medica, 36(18), 2465-2470.

Introduction must be extensive because it lacks references to numerous literature from recent years (both research and review articles)

In my opinion, it is necessary to include a summary table with the marked composition from the UPLC-MS study, and not the groups of compounds themselves. This should also be addressed in the discussion.

Please add information if the method was validated?

Fig 7 - no information about the concentration of which compound is described by the horizontal axis.

Ethanol concentration 70% (v:v)?

In my opinion, the article also needs a linguistic correction to make it easier to read.

Author Response

Manuscript ID: metabolites-2188193

Title: Metabolomics-based analysis of the effects of different cultivation
strategies on metabolites of Dendrobium officinale Kimura et Migo

Dear Reviewer ,

Thank you very much for your suggestions. We have carefully revised the manuscript according to your suggestions. Enclosed is the point-by-point response to these comments, along with the Highlights sections of the revised manuscript. We hope that with these reversions, this manuscript is suitable for publication in your journal. We thank the reviewers for their scientific perspectives.

I am looking forward to hearing from you sooner.

Sincerely yours,

Da Yang

Key Lab of the Basic Pharmacology of the Ministry of Education, Zunyi Medical
University,6 West Xue-Fu Road,Zunyi City, Guizhou

E-Mail: [email protected]

February 8, 2023

  1. It would be clearer to put the composition of verses 48-50 in the table with examples of specific compounds. Please write about previous composition studies

Response: We thank the reviewer for the comment. The compounds contained in D. officinale are more complex and are presented in the citations as classes of compounds. But for the flow and conciseness of the article, we have added a summary of the compounds mentioned in the citations to the “Supplementary material” in a table (Table s-2).

  1. Is it possible to show pictures of different cultivars of the plant?

Response: We thank the reviewer for the comment. We have added more sample plant information to the revised manuscript. Photos of the different cultivation methods have been added to the “Supplementary material”.

  1. Introduction must be extensive because it lacks references to numerous literature from recent years (both research and review articles).

Response: We thank the reviewer for the comment. We double-checked the literature and added more references in the "Introduction" section of the “revised manuscript”.

  1. In my opinion, it is necessary to include a summary table with the marked composition from the UPLC-MS study, and not the groups of compounds themselves. This should also be addressed in the discussion.
  2. Please add information if the method was validated?

Response: We thank the reviewer for the comment. We have a summary table with the marked composition from the UPLC-MS study, and QC verification by mixing samples. This information is displayed in “Supplementary material”. At the same time our analytical approach has been validated by previous studies and we have added references to the literature in the “revised manuscript”

  1. Fig 7 - no information about the concentration of which compound is described by the horizontal axis.

Response: We thank the reviewer for the comment. Regarding the antioxidant activity assay in Figure 7, we extracted D. officinale extracts from different cultivation methods by the same method to compare their antioxidant activity in the presence of the same amount of raw material.

  1. Ethanol concentration 70% (v:v)?

Response: We thank the reviewer for the comment. We thank the reviewer for the suggestion. We have revised the manuscript as suggested by the reviewer.

  1. In my opinion, the article also needs a linguistic correction to make it easier to read.

Response: We thank the reviewer for the comment. We tried our best to improve the manuscript and correcting mistakes. These changes will not influence the content and framework of the paper.

Reviewer 3 Report

The work is interesting but is only at the factual level with no hypotheses confirms that instrumentation can differentiate metabolites.   There is no clear evaluation of why the different growth methods influence metabolites- slight discussion that perhaps implies an effect of water deficit.   ALl of this should be scientifically examined.  

Methods on plant growth  inadequately detailed. Metabolic variation in plants is expected!  So  what tissues were examined  did these come from plants of the same developmental stage?  Flower formation, time of light period, presence of different  microbiomes  all could be involved in changing metabolic levels. 

Writing could be improved by editing.   Many sticky notes used at areas of concern

Author Response

Manuscript ID: metabolites-2188193

Title: Metabolomics-based analysis of the effects of different cultivation
strategies on metabolites of Dendrobium officinale Kimura et Migo

Dear Reviewer ,

Thank you very much for your suggestions. We have carefully revised the manuscript according to your suggestions. Enclosed is the point-by-point response to these comments, along with the Highlights sections of the revised manuscript. We hope that with these reversions, this manuscript is suitable for publication in your journal. We thank the reviewers for their scientific perspectives.

I am looking forward to hearing from you sooner.

Sincerely yours,

Da Yang

Key Lab of the Basic Pharmacology of the Ministry of Education, Zunyi Medical
University,6 West Xue-Fu Road,Zunyi City, Guizhou

E-Mail: [email protected]

February 8, 2023

  1. Methods on plant growth inadequately detailed. Metabolic variation in plants is expected! So what tissues were examined did these come from plants of the same developmental stage?  Flower formation, time of light period, presence of different microbiomes all could be involved in changing metabolic levels.

Response: We thank the reviewer for the comment. We have added sample plant information to the latest manuscript. Different cultivation methods are comprehensive influencing factor, so it is undeniable that the formation of flowers, light time and microorganisms have an impact on the metabolism of plants. This study were mainly compared the difference of metabolites of D. officinale under different cultivation methods, but even so, we still collected samples of D. officinale with the same flowering period, growth years and other important influencing factors under the condition of such comprehensive influencing factors. The research on light time and microorganisms needs more time and more systematic experiments, and we will carry out in the next research

  1. Writing could be improved by editing. Many sticky notes used at areas of concern

Response: We thank the reviewer for the comment. We tried our best to improve the manuscript and correcting mistakes. These changes will not influence the content and framework of the paper.

Round 2

Reviewer 2 Report

The Authors have improved their article according to suggestions. In my opinion it can be accepted in the present form.

Reviewer 3 Report

The authors need to use a professional editorial service for 

1) english grammar

2) organization

3) following the format of the journal 

Many comments are shown as sticky note for areas that detract from the quality of the paper

Not sure how measuring the neutral sugar residues as a measure of PS can possibly contribute to medicinal properties

The assessment of antioxidant potential is not complete

The paper essentially is a list of comparative facts   with no mechanistic inquiry